# Does parenting style moderate the relationship between parent-youth sexual risk communication and premarital sexual debut among in-school youth in Eswatini?

**Mduduzi Colani Shongwe**[1], **Min-Huey Chung**[2,3], **Li-Yin Chien**[4], **Pi-Chen Chang**[2]*

**1** Department of Midwifery Science, Faculty of Health Sciences, University of Eswatini, Mbabane, Eswatini, **2** School of Nursing, College of Nursing, Taipei Medical University, Taipei, Taiwan, **3** Department of Nursing, Taipei Medical University-Shuang Ho Hospital, New Taipei City, Taiwan, **4** Institute of Community Health Care, School of Nursing, National Yang-Ming University, Taipei, Taiwan

* pichen@tmu.edu.tw

## Abstract

### Background

Based on propositions of the contextual model of parenting style, we examined whether there is a relationship between parent-youth sexual risk communication (PYSRC) and premarital sexual debut, and whether this relationship is moderated by the parenting style.

### Methods

A cross-sectional study design was employed, and data were collected using a self-reported questionnaire from 462 youth (211 boys and 251 girls) aged 15–24 years in senior grades of three public high schools (two rural and one urban) in Eswatini (formerly Swaziland). A hierarchical binary regression analysis was conducted to examine the association between PYSRC and premarital sexual debut, and to test whether parenting style moderates this relationship.

### Results

The mean age of participants was 18.9 (±1.85) years, and a slight majority were females (54.3%). About 35.9% of participants reported having had sex (i.e., premarital sexual debut). After adjusting for age, gender, living arrangement, school location, and peer sexual activity, neither PYSRC (adjusted odds ratio [AOR] = 1.01, 95% confidence interval [CI]: 1.00, 1.03) nor parenting style (AOR = 0.81, 95% CI: 0.64, 1.04) significantly predicted premarital sexual debut in the sample. Likewise, parenting style did not significantly moderate the relationship between PYSRC and premarital sexual debut (AOR = 1.01, 95% CI: 1.00, 1.02).

**Data Availability Statement:** The authors confirm that, some access restrictions apply to the raw data underlying the findings. Data for the study were

collected with the approval of the Eswatini Health and Human Research Review Board (EHHRRB). Any researcher who intends to use these data must obtain approval from the EHHRRB. To request the data, an individual may either email the lead author (mduyaye@gmail.com) who will facilitate the request with the EHHRRB, or contact the EHRRB Secretariat directly at ehhrrbeswatini@gmail.com, or Eswatini Ministry of Health, P.O. Box 5, Mbabane, Eswatini. Otherwise, all pertinent data are within the paper and its Supporting information files.

**Funding:** The research did not directly receive any specific grant from funding agencies in the public, commercial, or not-for-profit sectors, the first author (MCS) conducted the study as part of his Master of Science studies which were sponsored by the Taiwanese Government through Taipei Medical University's International Student Type B Scholarship.

**Competing interests:** The authors have declared that no competing interests exist.

**Abbreviations:** AIDS, Acquired Immunodeficiency Syndrome; HIV, Human Immunodeficiency Virus; PSI-II, Parenting Style Inventory II scale; PTSRC-III, Parent-Teen Sexual Risk Communication III scale; PYSRC, Parent-Youth Sexual Risk Communication; SRH, Sexual and Reproductive Health; SSA, sub-Saharan Africa; STIs, Sexual Transmitted Infections.

## Conclusion

Contrary to propositions of the contextual model of parenting style, in this study, parenting style (authoritativeness) did not moderate the studied relationship, indicating the need for more studies to test the applicability of the contextual model of parenting style in African settings.

## Introduction

Globally, young people (i.e., those aged 10–24 years) are now reaching puberty earlier than in previous decades, and rates of early sexual debut are also rising or remain unchanged [1]. Ordinarily, simply engaging in premarital sexual intercourse is not a major health issue; however, it can be a precursor for risky sexual behavior later in life [2]. If no protection is used, premarital sex debut can result in serious adverse sexual health outcomes, such as an unwanted pregnancy, an abortion, regret, guilt, loss of self-respect, depression, loss of family support, substance abuse, and suicide [3]. Risk factors for premarital sex include being older; living alone or with friends; being in primary/elementary school or college; getting pocket money; having divorced or widowed parents [4, 5]; peer pressure; having strict parents; religiosity [6, 7]; watching pornographic material; substance abuse; being male [5, 8]; living in rural areas; and being in a romantic relationship [9]. Parent-child relationships; family environment; societal environment; cultural and traditional rules and values; economic conditions; and school environment are also some predictors of premarital sex debut [3].

In sub-Saharan Africa, efforts by national governments to control the spread of human immunodeficiency virus (HIV) among youth (i.e., those aged 15–24 years) have not yielded the desired results, as this group still accounts for two-thirds of all people living with HIV in this region [10]. Eswatini (formerly Swaziland), a small landlocked country in southern Africa with 1.1 million inhabitants, has not been spared this pandemic, as 20.9% of females and 4.2% of males aged 20–24 years are living with HIV [11], 94% of which was acquired through heterosexual intercourse [12]. Ironically, this occurs in a country where more than 90% of the population is Christian [13], a religion that strongly condemns premarital sex and regards it as immoral and sinful (fornication). In addition, Eswatini is one of the few countries in the world with a strong cultural heritage and highly conservative cultural norms and values, and traditions are highly safeguarded and widely practiced by a majority of the populace.

In Eswatini culture, premarital sex is strongly condemned, while chastity is encouraged and promoted through various traditional practices such as *Umhlanga* (Reed Dance), an annual national ceremony where unmarried girls who have no children assemble at the King's palace to celebrate their chastity. *Umcwasho*, a tassel worn on the head by unmarried girls who have no children during a prescribed period (usually lasting 1–5 years) when a girl is not supposed to be 'touched' by a boy, is an example of a similar cultural practice. Males also have their own similar annual cultural practice called *Lusekwane* (a national sacred shrub), whereby all unmarried boys who have no children are expected to assemble at the King's palace to receive the royal command to go and cut the sacred shrub during the annual *Incwala* ceremony [14]. These traditions are aimed at promoting chastity and therefore discouraging sex before marriage among the youth. However, despite the above-described religious rules and cultural customs, 54.5% of young women and 53.6% of young men aged 15–24 years reported having had sex [15], 3.5% of whom had sex before the age of 15 years [11]. Being cognizant of this problem, in 2014, the Eswatini government introduced a sex education curriculum in high schools

across the country; however, children do not receive sexual education at home, as talking about sexuality issues is taboo in many communities and households [16]. This is because parents fear that it may promote promiscuity and may serve as a license for adolescents to practice sex [17].

In her seminal work in the 1960s, Baumrind [18–20] conceptualized three parenting style typologies: authoritative (high demandingness and high responsiveness), authoritarian (high demandingness and low responsiveness), indulgent/permissive (low demandingness and high responsiveness). She established that authoritative parenting produced assertive, self-reliant children, whereas authoritarian parenting produced discounted, withdrawn children. On the other hand, permissive parenting produced children with low self-control and low self-reliance. Later, Maccoby and Martin [21] conceptualized parenting styles along two dimensions: responsiveness (the extent to which parents are warm, supportive, sensitive, interested, non-coercive, and accustomed to their children's needs) and demandingness (the degree to which parents demonstrate firm control, set maturity demands, provide supervision, enforce discipline, engage in direct confrontation, and establish performance or behavioural expectations for their children) [18, 19, 22]. Maccoby and Martin added a fourth parenting style: neglectful/uninvolved (low demandingness and low responsiveness) which is said to be associated with the worst child outcomes [21, 23].

In the early 90s, Darling and Steinberg proposed the contextual model of parenting style aimed at explaining the mechanism through which parenting practices influence child outcomes [24]. They defined parenting styles as constellations of attitudes that parents display towards their children that create emotional environments in which parents' behaviors are expressed [24]. The model postulated that parenting styles are different from parenting practices in that they consist of parent-child interactions across a wide range of situations, yet parenting practices are situation-specific and goal-directed behaviors that parents use to socialize their children in a particular setting [24]. The model envisions parenting style as a contextual variable that moderates the relationship between specific parenting practices and specific developmental outcomes. It hypothesizes that parenting styles alter the efficacy of parents' socialization efforts by buffering the effectiveness of particular practices, and by changing the child's openness to socialization. Therefore, the model does not take parenting style as a developmental process, but rather a characteristic of the parent and a feature of the child's social environment [24].

Research has shown that parenting styles have direct and indirect effects on children's outcomes [25], whereby adolescents of authoritative parents who have positive parental relationships, healthy open communication, and perceived parental support, are said to be less likely to engage in sexual risk behaviors [26]. Low levels of perceived parental warmth and parental knowledge have also been found to predict sexual onset [27]. To the contrary, using single item measures for each parenting style, Huebner and Howell [28], did not find direct effects of parenting style on sexual risk-taking.

Of all the parenting styles, authoritative parenting has been found to be the most effective style to child-rearing, as it leads to children who are socially responsible, achievement-oriented, and competent, with good decision-making abilities [23, 29]. However, there is paucity of studies testing the contextual model of parenting style. The few studies that have examined the relationship between parental practices and child outcomes, but not specifically testing the contextual model, have found that this relationship is strongest for children of authoritative parents [29–31]. Unfortunately, these studies have not adequately addressed the mechanisms through which parenting styles influence youth sexual behaviors.

Additionally, most research on parenting styles and parent-child processes has been conducted in developed countries, including in many ethnic societies (e.g., in African-American

communities) [32], but little has been done in the African region [33]. As a result, the mechanism through which parent practices and behaviors influence youth sexual behaviors is not well understood in this region, let alone in Eswatini. To try to understand these mechanisms, we adapted Darling and Steinberg's [24] contextual model of parenting style and examined whether parenting styles moderated the relationship between parent-youth sexual risk communication (PYSRC) and premarital sexual debut among in-school youth in Eswatini. The contextual model was used as a foundation for formulating the research questions, as well as for the hypothesized directions in our conceptual framework (see S1 Fig). We hypothesized that parenting style would moderate the relationship between parent-youth sexual risk communication (parental practice) and engagement of the youth in premarital sex (child outcome).

## Methods

### Study design, setting, and sampling procedures

A cross-sectional design, using a well-structured, self-administered questionnaire was employed, and data were collected in August 2012 from in-school youth aged 15–24 years in Forms 4 and 5 (Grades 11 and 12) of three public high schools (two rural and one urban) in the Manzini region of Eswatini. This region was chosen because it has the highest number of public high schools and in turn enrolls the greatest number of high school students. In order for a student to be part of this study, they had to be either in Form 4 or 5 (Grades 11 or 12); be a public high school pupil; be a daily scholar; be aged 15–24 years; and have regular interactions with at least one parent or guardian. Pupils who lived in an orphanage, boarding school, a private school, gender-segregated school, and/or who reported to have no interaction with at least one parent or guardian were excluded from the study.

A two-stage stratified cluster sampling technique was used to randomly select schools from two separate strata. In the first stage, the schools were listed separately by urban and rural strata, and thereafter the name of each school was written on pieces of paper of equal sizes and of same color, folded and put in a box which was then shaken to shuffle the papers thoroughly [34]. A colleague of the first author who was not involved in the study in anyway was asked to pick two papers in succession for the rural schools and one for urban schools, following a reshuffle with each selection session. The number of schools per strata was determined by the distribution of the general population in Eswatini (70% rural) and the minimum sample size needed as per the sample size calculation. In the second stage, in each grade, cluster sampling was employed, whereby all pupils in Forms 4 and 5 (Grades 11 and 12) in the selected schools were eligible to be included in the study and were therefore invited to participate (see S2 Fig). Therefore, the classrooms were the sampling unit in this study.

### Sample size determination

The sample size was calculated using OpenEpi version. 2.3.1 [35], based on the 2011 estimate of 83,096 pupils attending secondary and high schools in Eswatini [36]. At a 95% confidence level, with a hypothesized frequency outcome factor of 50% (which yields the largest sample size), and a design effect of 1 (for a random sample), the minimum desired sample size was 383. However, to increase the statistical power, we recruited all students in the selected schools, and after data cleaning, 462 usable questionnaires were retained for data analysis.

### Data collection process

Permission was requested and granted by school principals and class teachers to utilize classes as venues for students to fill in the questionnaire. Lessons were not disturbed during the time

of data collection, because students were no longer having lessons as it was the end of the second term. Data collection took two days in each school. On the first day, the study information sheet and informed consent forms were distributed to students who were aged 15–17 years to request their parents' informed consent for them to participate in the study. On the second day, questionnaires were distributed in sealed envelopes to students who returned signed consent forms and those who were 18 years or older while they were seated in their regular class seats. Filling in the questionnaire took each participant about 30 minutes on average.

## Measures

**Independent variable.** PYSRC was assessed using the Parent-Teen Sexual Risk Communication Scale (PTSRC-III) [37]. The PTSRC-III measures the amount of sexual risk communication adolescents had with each of their parents in eight specific topical areas. Response choices are scored from 1 (none) to 5 (extensive) for the amount of communication on each specific topic. Scores for mothers' and fathers' subscales were computed separately by summing the item scores and combining them into a single measure of a total PTSRC-III score that ranged from 16–80, with a higher score reflecting more-extensive parent-youth communication. The reliability of the PTSRC-III has been well established [38–40]. In this study, the PTSRC-III showed good internal reliability with an overall Cronbach's alpha of 0.90.

**Dependent variable.** Premarital sexual debut: Participants had to report whether they had ever had sexual intercourse (yes = 1) or not (no = 0) in their lifetime.

**Moderator.** Parenting style was measured using the Parenting Style Inventory-II (PSI-II) [41] which identifies participants' general experiences with their mothers and/or fathers. The original instrument contains three subscales (with five items each) measuring emotional responsiveness, demandingness, and autonomy-granting parenting. Respondents rated the extent to which they agreed with each item, for both their mothers and fathers, using a 7-point scale (1 = strongly disagree to 7 = strongly agree), making the highest possible score on each dimension 35. However, the autonomy-granting dimension was excluded in the main analysis testing our hypothesis in this study, consistent with previous studies [21, 42], because it tends to be highly correlated with the other two dimensions combined. Instead, we combined the item mean scores of both parents' responsiveness and demandingness subscales to create one measure called authoritative parenting style, with a higher score indicating higher authoritativeness. The advantage of this technique is that it does not exclude youth with only one parent, and helps to avoid collinearity and power problems associated with simultaneously entering inter-correlated subscales [43].

In our study, intercorrelations between the dimensions were as follows: demandingness with responsiveness, $r = .76$; demandingness with autonomy-granting, $r = -.14$; and responsiveness with autonomy granting, $r = -.13$. The high collinearity between the demandingness and responsive dimensions was not of concern since we combined these two dimensions to form one parenting style (authoritativeness) [42]. In line with previous studies [44, 45], we report the item mean score (1–7) instead of the total parenting style mean score (10–70). Cronbach's alpha of the authoritativeness scale in this study was 0.73.

Perhaps, noteworthy are the inconsistencies in measurement and scoring of parenting style in parenting style literature [46]. The most common method is classifying individuals into subgroups by using cutoff-points based on tertile split, median-split, or by use of one-item measures to come up with four typological parenting styles [47, 48]. However, Lanza, et al. [49] observed that imposing arbitrary cut-off points makes it difficult to compare across samples and to generalize findings. It also tends to exclude some participants from the analyses e.g. those who do not meet the criteria for predetermined typologies, which may, however,

decrease a study's ability to accurately predict outcomes or to identify individuals who could benefit most from prevention or intervention efforts [49].

Another approach is by directly examining the main constituent dimensions of the parents' behaviours towards their children [21, 41, 49, 50]. Odubote [51] noted that the advantage of using the "dimensional approach" is that it enables researchers to identify the unique effects of each dimension of parenting style on child behaviours, while the typological approach uses a combination of all parenting style dimensions, or uses the interaction effects of the dimensions (see Stewart and Bond [46] for a thorough discussion about the inconsistencies and lack of standardization of measures in the study of parenting). At the time of data collection for this study, there was no published study on parenting styles in Eswatini, therefore, we did not know which parenting style was optimal in the country, hence, we decided to test one of the cross-cultural, well-established effective parenting style [22, 50], (authoritativeness) used largely by Darling and Steinberg in their multiple works on parenting style. We used the "dimensional approach" only for the post-hoc analyses.

**Control variables.** All of the covariates were identified during a literature review. We had collected a number of control variables from the participants, but not all were controlled for, as it depended on how the model performed with each variable added into the model. We judged the contribution of each variable in the model based on its Wald statistic value, clinical relevance, and its $p$ value. As a result, in all blocks of the multivariate models, we controlled for age, gender, area of residence (rural and urban), living arrangement, and peer sexual activity (a proxy for peer pressure). We excluded 'Form/Grade' as it was highly correlated with 'age'.

## Data analysis

Data were analyzed using IBM Statistical Package for Social Sciences (SPSS) for Windows version 25.0 [52]. In the bivariate analysis, we performed Chi-square and $t$-tests to compare sociodemographic characteristics between participants who had engaged in premarital sex and those who had not. The regression analysis and interpretation of the results followed guidelines suggested by Frazier et al. [53] and Bennett [54]. A four-step hierarchical logistic regression was conducted i.e., variables were entered in blocks in SPSS (see S1 File), whereby the interaction term was entered in the final step to control for Type I error [53]. We retained the non-significant interaction term in our final model as we had a strong theoretical basis to expect moderator effects [55]. Since our interaction term was not significant, we further conducted exploratory post-hoc analyses using SPSS PROCESS macro [56], stratified by parents' gender as well as gender and age of the youth to ascertain if there were no underlying main and moderation effects within each of these variables [57]. We dichotomized age of the youth as15-17 years and 18–24 years post-hoc analyses to test potential moderation effects of the different parenting style dimensions in the studied relationship. The age cut-off points were informed by that the age of sexual consent is 18 years in Eswatini. Age has also been reported as one of the moderators in associations between parenting practices and child outcomes [57]. We fitted models for all the possible moderator effects, first with both sexes of the parents, and thereafter, for each sex of the parent i.e. mother's or fathers' sexual risk communication (independent variable) with premarital sex (the dependent variable) and mothers' or fathers' authoritativeness, demandingness, responsiveness and autonomy-granting (the potential moderator), separately for each age group (S2 and S3 Files). For all inferential tests, the level of alpha was set at 0.05.

## Ethical considerations

Ethical clearance (Ref: MH/599C) was granted by the then Scientific and Ethics Committee of the Eswatini Ministry of Health. All participants ($\geq$ 18 years) who agreed to participate in the

study had to sign an informed consent form, while parental written consent and assent (from the students) were obtained for participants aged less than 18 years. To ensure confidentiality and anonymity, the teachers of each school were not part of this study and were not present in the classrooms when participants filled in the questionnaires.

## Results

### Background characteristics

As shown in Table 1, a majority (54.3%, $n = 251$) of participants were female, enrolled in urban schools (53.5%, $n = 247$), were unsure as to the number of friends who had already engaged in premarital sex (46.3%, $n = 214$), and 53% ($n = 245$) reported to have regular interactions with both of their parents. Nearly 36% (n = 166) of the participants had had premarital sex, a majority of whom did so at 17 years or older (54.8%, $n = 91$). In the bivariate analysis, participants who had engaged in premarital sexual significantly differed from those who had not by gender ($p < 0.001$), school location ($p = 0.008$), grade ($p = 0.005$), peer sexual activity ($p < 0.001$), most influential parent's education level ($p = 0.04$), and age ($p < 0.001$). However, there were no statistically significant differences in the mean scores of PYSRC ($p = 0.49$ and of authoritativeness ($p = 0.15$) between the two groups (Table 1).

### Multiple logistic regression results

Interpretation and reporting of the main effects in this study followed recommendations by Frazier and colleagues [53]. After adjusting for other covariates in step 4 (model 4), neither PYSRC (AOR = 1.02, 95% CI: 0.994, 1.039) nor authoritativeness (AOR = 0.82, 95% CI: 0.643, 1.050) significantly predicted youth premarital sexual debut, when authoritativeness score is conditioned at 0 in each case. Similarly, the interaction term (PYSRC x Authoritativeness) showed no statistically significant moderating effect on the association between PYSRC and premarital sexual debut (AOR = 1.01, 95% CI: 0.995, 1.022), as shown in Table 2. However, it is worth mentioning that in model 3 (i.e., after adding authoritativeness), PYSRC became statistically significant, (AOR = 1.02, 95% CI: 1.001, 1.042, $p = 0.044$)], but the borderline significance did not hold in model 4 when the interaction term was added (Table 2).

Results from the exploratory post-hoc analyses stratified by gender and age of the youth did not reveal any significant moderating effects for all models (see S2 File); hence we report results from the combined model. As recommended by Frazier and colleagues [53], moderation effects were not plotted for the results of the main hypothesis test in this study since they were not statistically significant. We further explored if parents' gender did not suppress the effects of each parenting dimension, however, we still did not find statistically significant effects except for mothers' demandingness which was found to significantly moderate the association between mothers' sexual-risk communication and premarital sex, $p = 0.04$. Since these results were conducted as part of the post-hoc exploratory analyses, the simple slopes of the moderation effects are not presented here, but are included as S3 File.

## Discussion

Contrary to propositions of the contextual model of parenting style, we found no statistically significant moderating effects of parenting style, nor did we find statistically significant associations of PYSRC or parenting style with premarital sexual debut. In the bivariate analysis, the finding that a higher proportion of males had engaged in premarital sex than females is consistent with the literature [8, 58]. Similarly, there was a high proportion of participants who had engaged in premarital sex among those whose peers had already engaged in premarital sexual

**Table 1. Characteristics of participants by premarital sexual debut.**

| Variable | Total | Premarital sex | Never had sex | $X^2$ | p value |
|---|---|---|---|---|---|
| | (N = 462) | (n = 166) | (n = 296) | | |
| | n (%) | n (%) | n (%) | | |
| **Gender** | | | | | |
| Female | 251 (54.3) | 65 (39.2) | 186 (62.8) | 24.04 | < 0.001 |
| Male | 211 (45.7) | 101 (60.8) | 110 (37.2) | | |
| **School location** | | | | 7.14 | 0.008 |
| Rural | 215 (46.5) | 91 (54.8) | 124 (41.9) | | |
| Urban | 247 (53.5) | 75 (45.2) | 172 (58.1) | | |
| **Grade** | | | | 8.06 | 0.005 |
| Form 4 | 263 (56.9) | 80 (48.2) | 183 (61.8) | | |
| Form 5 | 199 (43.1) | 86 (51.8) | 113 (38.2) | | |
| **Living arrangement** | | | | 1.85 | 0.40 |
| Both biological parents | 169 (36.6) | 54 (32.5) | 115 (38.9) | | |
| Single parent | 169 (36.6) | 64 (38.6) | 105 (35.5) | | |
| Guardian/Other | 124 (26.9) | 48 (28.9) | 76 (25.7) | | |
| **Peer sexual activity** | | | | 49.04 | < 0.001 |
| Not sure | 214 (46.3) | 66 (39.8) | 148 (50.0) | | |
| None/few friends | 157 (34.0) | 39 (23.5) | 118 (39.9) | | |
| Half/most of friends | 91 (19.7) | 61 (36.7) | 30 (10.1) | | |
| **Parents' education¥** | | | | 8.13 | 0.04 |
| Preschool/none | 24 (5.2) | 15 (9.0) | 9 (3.0) | | |
| Primary school | 54 (11.7) | 17 (10.2) | 37 (2.5) | | |
| Secondary/High school | 234 (50.6) | 80 (48.2) | 154 (52.0) | | |
| Tertiary | 150 (32.5) | 54 (32.5) | 96 (32.4) | | |
| **Regular interactions** | | | | 3.54 | 0.17 |
| with mothers only | 181 (39.2) | 65 (39.2) | 116 (39.2) | | |
| with fathers only | 36 (7.8) | 18 (10.8) | 18 (6.1) | | |
| with both parents | 245 (53.0) | 83 (50.0) | 162 (54.7) | | |
| **Age at sexual debut** | | | | N/A | |
| ≤11–16 years | - | 75 (45.2) | | | |
| ≥17 years | - | 91 (54.8) | | | |
| | **Mean (SD)** | | | t | |
| Age in years | 18.9 (1.85) | 19.70 (1.77) | 18.45 (1.74) | -7.40 | < 0.001 |
| PYSRCᵃ | 30.89 (14.32) | 31.51 (15.44) | 30.54 (8.97) | -0.70 | 0.49 |
| Authoritativeness | 3.89 (1.36) | 3.77 (1.39) | 3.96 (1.34) | 1.45 | 0.15 |

**Notes**: All participants were Christian, unmarried, and of black ethnicity.

¥ Most influential parent's education (defined in the questionnaire as: "the one whom you look up to for advice and whom you are closer to or relate better with").

ᵃ PYSRC, parent-youth sexual risk communication; SD, Standard deviation.

activity (peer pressure), similar to other studies [6, 7] and earlier findings from Eswatini [16]. We also found that youth who had engaged in premarital sex were older than those who had not, consistent with findings by Waktole [5] and Young et al. [59]. This is not surprising since as youth grow older, they reach the age of sexual consent and therefore may feel more at ease initiating sexual activity.

Participants in this study reported lower parent-youth communication than in studies conducted in the United States [39, 40]. This could have been due to a majority of participants in

**Table 2. Hierarchical logistic regression model testing moderation effects (N = 462).**

| Model [a] | B (SE) | AOR (95% CI) | p value |
|---|---|---|---|
| **Step 2: Model 2** | | | |
| PYSRC | 0.01 (0.01) | 1.01 (0.995, 1.027) | 0.199 |
| **Step 3: Model 3** | | | |
| PYSRC | 0.02 (0.01) | 1.02 (1.001, 1.042) | 0.044 |
| Authoritativeness | -0.21 (0.12) | 0.81 (0.638, 1.038) | 0.097 |
| **Step 4: Model 4** | | | |
| PYSRC | 0.02 (0.01) | 1.02 (0.994, 1.039) | 0.150 |
| Authoritativeness | -0.20 (0.13) | 0.82 (0.643, 1.050) | 0.117 |
| PYSRC x Authoritativeness | 0.01 (0.07) | 1.01 (0.995, 1.022) | 0.242 |

Notes:

[a]Step 1 included control variables only, therefore, all models were adjusted for residence, peer sexual activity, gender, age, and living arrangement; All continuous variables were centered.

**Abbreviations**: PYSRC, parent-youth sexual risk communication; AOR, adjusted odds ratio; CI, confidence interval; B, unstandardized coefficient, SE, standard error.

our sample reporting largely having interactions with their mothers, and few having interactions with their fathers, and when summing up the scores, the overall PYSRC score was reduced. However, it is also possible that in reality, parents in Eswatini could be communicating less with their children about sexuality issues, since traditionally, this kind of communication is regarded as taboo, and instead, youth might be obtaining information about sexuality from elsewhere, such as from peers [16].

In the multivariate analysis, the absence of significant associations between PYSRC, or parenting style with premarital sexual debut suggests that other variables, which we did not measure in our study (confounders), might account for most of the variance in premarital sexual debut among youth in Eswatini, and that PYSRC and parenting styles are the least of them [31]. For example, Somers and Paulson [60] found that higher parental closeness in conjunction with child communication was unrelated to the sexual behavior of children, and instead, age was a stronger predictor. Therefore, in our case, it is also important to consider whether parenting style alone is adequate to explain child outcomes across populations and in different settings, or if it needs to be supplemented with alternative factors for some groups and not others [61]. For that reason, we cannot rule out that possibility in our study due to confounding. In fact, the borderline significant association between PYSRC and premarital sexual debut (see Table 2, model 3) suggests that some of the variance in premarital sexual debut is explained by a combination of PYSRC and parenting style, which suggests that PYSRC alone might be inadequate to explain the variance in premarital sexual debut, but when put together with authoritativeness, it explained some of the variance in premarital sexual debut.

However, we should point out that the direction of the borderline significant association (see text of multivariate results) is not what we had expected, as it showed that the greater the amount of PYSRC, the higher the odds of engaging in premarital sex. One possible explanation for this finding is that once parents notice that their children are dating or suspect that they are sexually active, they react by initiating or increasing discussions with their adolescents about sexual risks. Alternatively, it could also be that once the youth become sexually active, they approach their parents to seek clarity on certain topics related to their sexuality and therefore are the ones who initiate the discussions. However, the latter scenario is less likely in Eswatini as sexuality education is regarded as taboo, hence Swati children may be afraid to

initiate conversations about sexuality topics with their parents. Unfortunately, due to the cross-sectional nature of the study, we could not establish the temporality between PYSRC and premarital sexual debut. The PTSRC-III also does not ask participants when sexual risk communication began.

We also did not find evidence of moderating effects of parenting style on the relationship between PYSRC and premarital sexual debut, despite the item mean score for the PSI-II being comparable with mean scores of previous studies [43, 45]. To the contrary, studies in other domains [62, 63] found significant moderating effects of parenting styles on child outcomes, while Schary and colleagues [64] found no such effects. One likely explanation for the absence of significant moderator effects in this study could be that perhaps parenting styles also act in other pathways other than those proposed by the contextual model of parenting style. For example, Li and colleagues [65] found that parenting styles mediate parent-child relationships. In light of these conflicting findings, there is a need for more studies using larger samples and longitudinal designs to further examine the mechanism through which these parental processes influence child outcomes.

It is also possible that the absence of significant main and moderation effects in this study could have been due to the cultural context of parenting styles and parent-child processes in Eswatini vis-à-vis other settings such as in Western societies, where much of the mainstream literature on parenting has been published. A recent qualitative study among adolescent in Tanzania found that most parents appeared to use an authoritarian style of parenting [66], suggesting cultural differences in parenting style in different context. Mainstream thinking and research about parent-child relationships have been dominated by Western cultural beliefs, which may have erroneously assumed that the meaning of parenting is similar across cultures [67]. However, such ideology may obscure important differences in what various cultures expect of and understand about parenting and parent-child relationships, especially because parenting practices vary from one culture to the next [68].

Phoenix and Husain [61] made strong arguments about bias in generalizing parenting styles across cultures and ethnicities. They argued that child outcomes may be affected by contextual or cultural factors, and that each culture has different goals and expectations of their citizens, such that, an effective parenting style elsewhere (e.g., authoritative parenting style in the United States of America) might not be as effective in other cultures [61]. This view is in line with an earlier study by Chao [69] who found that constructs of authoritative and authoritarian parenting were not relevant to Chinese-Americans. A recent systematic review of 428 studies on parenting styles and child outcomes also found that in western countries, associations of authoritarian parenting with academic achievement were less negative in Hispanic families than in non-Hispanic, White families. In all regions of the globe (with some regional variation), authoritative parenting was found to be associated with at least one positive child outcome, whereas authoritarian parenting was found to be associated with at least one negative outcome [70].

Parenting styles are also said to vary by geographical location; religious beliefs; parental culture; between and within individual parents; family and personal values; personality and temperament of the parents and child; socioeconomic status; and ethnicity [71]. Therefore, in this study, we could not rule out the possibility that global parenting styles might not account for premarital sexual debut of youth in Eswatini, and instead, other factors that interact with parental practices may be responsible for youth premarital sexual debut. For example, a previous study [72] found that subjective norms were the strongest predictors of intention for premarital sexual abstinence among in-school youth in Eswatini. This was also observed in other settings, where norms; values; social class; parental attitudes and approval; gender; and ethnicity were found to produce outcomes that ran counter to parenting style effects on child

outcomes [61, 73]. Future studies should therefore investigate the influence of such factors on sexual behaviors of youth.

In exploratory post-hoc analyses, none of the youth-age or gender-stratified analyses produced any significant main or moderating effects. One might question whether sexual behavior in the mid-20's is less related to parenting variables than sexual behaviors in teens, which might have contributed to the null result of the main hypothesis test in our study. We cross-tabulated the current age of the youth against their age at sexual debut to ascertain the proportion of adults who had their sexual debut in adulthood versus in childhood. Nevertheless, the cross-tabulation revealed that nearly 60% of older youth (18~24 years of age) had their sexual debut in adulthood. Having said that, we should point out the limitation in the measurement of the age-at-sexual-debut variable: participants were asked to choose from predetermined categories of 11 or younger, 12~16, and 17 or older, instead of 18 years or older. Unfortunately, that is how the variable is structured in the YSRBS which is the instrument we adopted for our study, and it was deemed appropriate at the time of data collection since the age of sexual consent in Eswatini was 16 years back then.

We further performed independent-sample *t*-tests to ascertain if parenting style and parent-youth sexual risk communication significantly differed by the current age of the youth and their age at sexual debut, and found no significant mean differences. In addition, since none of the youth-age or gender-stratified exploratory analyses showed statistically significant effects, it does not seem like the older youth in the sample suppressed the effect of parenting variables in the younger sample, which would have been the case had we found significant effects only in the younger sample (<18 years of age). In summary, as much as it is plausible that sexual behavior in older youth is less likely to be related to parenting variables than in younger youth, that does not seem to be the case in the current sample. One plausible reason could be that Swati parents generally do not talk to their children about sexuality issues (as discussed in detail above in the preceding text), independent of the stage or age of the child, such that the slope is constant across the adolescent and youth life course. Outputs from the cross-tabulations and *t*-tests are presented in the S4 File.

The only post-hoc analysis result of a significant moderating effect was found when parenting style dimensions were separately included in the models. The finding of simple slopes means that the effect of mother-youth sexual risk communication on premarital sex differed at different levels of demandingness, such that for low mothers' demandingness scores (i.e., those below the mean), mother-youth sexual risk communication did not predict premarital sex, whereas it significantly positively predicted premarital sex at the mean level and at high mother demandingness (above the mean) scores (see S3 File). Again, the lack of information on the timing of communication between the mothers and their children limits our full interpretation of this finding as reverse causation is a possibility, as discussed above. More studies that include parent-child dyads are needed to further investigate this association separately based of the gender of the parents and youth.

## Strengths and limitations

This study is among a few studies that reported data on both sexes of parents and of the youth which enhances the interpretation of the results and enhances the study's internal validity. The study obtained information from the perspective of the youth as consumers of parent-child communication, which may help promote acceptability of the findings to the youth and further inform youth sexual and reproductive health programming in the direction that the youth desires. To our knowledge, this is the first study to test the contextual model of parenting style in Eswatini and in the region, and therefore contributes to the

body of knowledge and theory advancement with regards to parent-child processes especially in southern Africa.

Despite the strengths discussed above, the study still has a few limitations. First, all of the responses were based on self-reports and therefore were not immune to reporting bias. To minimize this bias, we ensured that respondents recorded their responses while seated in an examination set-up such that the person sitting next to each participant could not see the other person's responses. We further provided envelopes which they used to shield their responses and to enclose the questionnaire once they had finished. Teachers were also not present when the students filled out the questionnaires, and therefore we believe that there was no external pressure which might have influenced their responses. Respondents were also reassured that their responses would not be shared with their teachers.

Second, we could not completely rule out social desirability bias. However, respondents were assured of anonymity and confidentiality and the importance of being honest when answering the questions as a measure to minimize this bias. Prior to filling in the questionnaire, we also explained to the respondents that some of the questions depended on the responses of other questions in the questionnaire, so that it would be easy for researchers to note any incongruence in the responses. The questionnaire was also structured in such a way that there were no 'skip' questions, in order to minimize cases where participants could decide to answer 'no' to a certain question, so that they could jump to the end of the questionnaire. That also ensured that we had complete data for all questions.

Third, this was a cross-sectional survey, and therefore this study could neither change the behaviors of the youth nor establish causality among the studied associations. Fourth, participants were asked to recall past experiences which may have introduced some recall bias. Fifth, the study also assessed parenting styles and the amount of sexual risk communication as perceived by the youth and therefore lacked data from their parents' perspective, who might have reported either similar or different parenting information if asked. Lastly, we did not collect data from out-of-school youth, and therefore the findings might not accurately represent the conditions for all youth in Eswatini. As such, caution should be exercised when generalizing the study findings to other settings due to cultural and contextual differences.

## Conclusions

This study is among the first to empirically test the contextual model of parenting style in a sub-Saharan African sample. Therefore, our findings contribute to the growing body of knowledge on parenting styles and child outcomes. We found no evidence of an association between PYSRC and premarital sexual initiation, and no evidence of moderating effects of parenting style between these two variables. In light of these findings and the study limitations, longitudinal studies are warranted to better understand the mechanisms through which parental practices and parenting styles influence youth sexual behaviors, especially in Africa.

## Definitions of SiSwati words [14, 74]

*Incwala*: also known as the Kingship Ceremony, is an annual national event to commemorate the end of the year and the beginning of a new one, as a way of paying homage to the country's ancestors.

*Lusekwane*:–a sacred shrub picked by young boys during the *Incwala* ceremony and also as part of a custom to preserve chastity in boys.

*Umcwasho*: a tassel worn by girls during a prescribed national campaign as part of a custom to preserve chastity in boys and girls.

*Umhlanga*: a reed picked by young girls as part of a custom to preserve chastity in girls. The annual ceremony for picking the reed is also known as *Umhlanga* (Reed Dance Ceremony).

## Supporting information

**S1 Fig. Conceptual framework.**
(DOC)

**S2 Fig. Schematic view of the sampling procedure.**
(DOC)

**S1 File. SPSS output file.** Hierarchical logistic regression analysis by gender and age.
(SPV)

**S2 File. PROCESS macro output file.** Post-hoc analysis for parenting style by gender and age.
(SPV)

**S3 File. PROCESS macro output file.** Post-hoc analysis for statistically significant moderation effects.
(SPV)

**S4 File. SPSS output file.** Cross-tabulations and t-tests.
(SPV)

## Acknowledgments

We would like to extend our sincere gratitude to the students (and their parents) who agreed to participate in this study, and to the school managers for the administrative permission to enter their schools. We would also like thank Prof. Katherine M. Hutchinson and Prof. Nancy Darling for permission to use the PTSRC III and PS-II scales, respectively. We further thank Prof. Kuei-Ru Chou for her technical guidance at the conceptualization stage of the study, Dr. Yen-Kuang Lin for statistical guidance, and Lindelwa Portia Dlamini for her constructive comments on an early draft of the manuscript.

## Author Contributions

**Conceptualization:** Mduduzi Colani Shongwe, Pi-Chen Chang.

**Formal analysis:** Mduduzi Colani Shongwe.

**Investigation:** Mduduzi Colani Shongwe.

**Methodology:** Mduduzi Colani Shongwe, Min-Huey Chung, Li-Yin Chien, Pi-Chen Chang.

**Supervision:** Pi-Chen Chang.

**Writing – original draft:** Mduduzi Colani Shongwe.

**Writing – review & editing:** Min-Huey Chung, Li-Yin Chien, Pi-Chen Chang.

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
