## [Decision Letter · Decision Letter 0]

31 Jul 2020

PONE-D-19-35873

Does parenting style moderate the relationship between parent-youth sexual risk communication and premarital sexual debut among in-school youth in Eswatini?

PLOS ONE

Dear Dr. Chang,

Thank you for submitting your manuscript to PLOS ONE. After careful consideration, we feel that it has merit but does not fully meet PLOS ONE’s publication criteria as it currently stands. Therefore, we invite you to submit a revised version of the manuscript that addresses the points raised during the review process.

Please accept my apologies for the misunderstanding regarding the overlap with previous work - I understand now that the previous paper reviewer 2 refers to is a masters thesis, so we are not concerned about this aspect of the manuscript anymore. Please do take a look at the methodological and other points raised by both reviewers when preparing your revised manuscript.

We look forward to receiving your revised manuscript.

Kind regards,

Dr Joseph Donlan

Staff Editor

PLOS ONE

Journal Requirements:

a) Did parents provide their written or verbal informed consent for minors to participate in this study?

4. We note you have included a table to which you do not refer in the text of your manuscript. Please ensure that you refer to Table 2 in your text; if accepted, production will need this reference to link the reader to the Table.

5. Please upload a copy of Supporting Information S3 and S4 Output files which you refer to in your text on page 25.

Reviewers' comments:

Reviewer's Responses to Questions

**Comments to the Author**

1. Is the manuscript technically sound, and do the data support the conclusions?

Reviewer #1: Yes

Reviewer #2: Partly

2. Has the statistical analysis been performed appropriately and rigorously? 

Reviewer #1: Yes

Reviewer #2: Yes

3. Have the authors made all data underlying the findings in their manuscript fully available?

Reviewer #1: Yes

Reviewer #2: Yes

4. Is the manuscript presented in an intelligible fashion and written in standard English?

Reviewer #1: Yes

Reviewer #2: Yes

5. Review Comments to the Author

Reviewer #1: I highly commend the authors for conducting a research study that tests the construct of parenting style as a moderator of the relationship between parent-youth sexual risk communication (PYSRC) and premarital sexual debut. Moreover, results from the study further confirm that the construct of parenting style should be considered within cultural context, with attention given to global variability in application or lack thereof. However, based on Baumrind's conceptualization of parenting styles, I would suggest incorporating the other dimensions of authoritarian and permissive (indulgent, neglectful) in investigating the moderating effects of parenting style given the relationship between parenting practices and child development.

Regarding the introduction, more detail could be given on Darling and Steinberg's contextual model of parenting style; however, Baumrind's conceptualizations of parenting style (1971, 1991) should also be noted as part of the theoretical framework. A more thorough review of literature should be presented, highlighting relevant studies that inform the current research and identify significant gaps.

Concerning the methodology, the sampling procedures seem to lack academic rigor. The use of random drawing from slips of paper appears to be devoid of sophistication worthy of scientific, empirical investigation. Also, more accuracy and detail is needed on the cluster sampling of each grade and the procedure by which individual random sampling is accomplished with each classroom. Two-stage cluster sampling appears to be an appropriate, but accuracy in the application of such a methodological design is lacking. Lastly, you should clarify for the readers what is meant by Form 4 and 5.

For the discussion, in Lines 270, 277, 318 and 351, the names of authors should be mentioned instead of "studies elsewhere" or "other studies."

Regarding grammatical errors, there are some run-on sentences and lack of appropriately placed semicolons. For example, in Lines 59-63, there are run-on sentences and a need to use semicolons for separation. In Lines 63-66 and Lines 345-347, semicolons are also needed to separate long lists.

In Lines 117-118, African-American communities should be placed in parenthesis.

In Line 278, few "had" interactions should be replaced by few "having" interactions.

In Line 337, bias should be used instead of biasness.

Reviewer #2: Review of: Does parenting style moderate the relationship between parent-youth sexual risk communication and premarital sexual debut among in-school youth in Eswatini?

(The paper presents the results of a survey done in 2012. Unfortunately, the paper is very similar to one published in 2013 using the same data which renders the contribution questionable, but more importantly, fails to meet ethical standards. I’m not able to determine the degree of self-plagarism, but at the least, substantial p-hacking appears to have occurred.)

There are other issues, as well. The authors present the findings as related to “parent-youth sexual risk communication”; however, the sample is comprised of individuals ranging in age up to 24, averaging nearly 19. The sample includes adults and adolescents. Because the measure of sex is whether or not they have ever had sex rather than voluntary debut and age of debut, the question of whether a parent has influence has to be considered. At the least, the sample should be divided into minors and adults for comparisons before aggregating but a better approach would have been to do that AND use age of debut as the dv.

The findings indicate that authoritativeness did not predict sexual debut. The choice of how to manage parental style was based on Darling’s treatment of her scale; however, you have the ability to determine whether, based on your sample, that was appropriate. Having used Baumrind’s scales in my own work, there is no limitation of the scale in one-parent households and the collinearity problem seems more likely to be an issue with Darling’s scale but that is something the authors can and should test for themselves.

The authors also have the ability to conduct post-hoc tests regarding whether the mother or father has more or less influence.

The lower PYSRC scores suggest why authoritativeness has little influence and indicate the need to examine that construct cross-culturally. The lower scores also may be the result of the adult population in which this study was conducted (see my earlier concern).

The topic of study is important and I wish I could be more positive about this paper. But I do hope the authors will collect new data and address the issues I have raised regarding cultural differences in parental style, age of debut as the dv, and the like.

6. PLOS authors have the option to publish the peer review history of their article (what does this mean?). If published, this will include your full peer review and any attached files.

Reviewer #1: **Yes: **Jewrell Rivers

Reviewer #2: No

---

## [Author Response · Author response to Decision Letter 0]

17 Sep 2020

PONE-D-19-35873

Does parenting style moderate the relationship between parent-youth sexual risk communication and premarital sexual debut among in-school youth in Eswatini?

Dear Dr Joseph Donlan:

Thank you for your ongoing consideration of our manuscript (PONE-D-19-35873) for publication in the PLOS ONE. We appreciate the time spent by you and the reviewer and believe the revised manuscript is improved. We have address the reviewers’ comments one by one.

We look forward to hearing from you regarding our submission. We would be glad to respond to any further questions and comments that you may have.

Sincerely,

Pi-Chen Chang, RN., PhD. 

School of Nursing, College of Nursing, Taipei Medical University

250 Wu Hsing Street, Taipei City, Taiwan, 110 R.O.C.

Tel. +886-2-2736-1661

Email: pichen@tmu.edu.tw

---

## [Decision Letter · Decision Letter 1]

16 Dec 2020

PONE-D-19-35873R1

Does parenting style moderate the relationship between parent-youth sexual risk communication and premarital sexual debut among in-school youth in Eswatini?

PLOS ONE

Dear Dr. Chang,

Thank you for submitting your manuscript to PLOS ONE. After careful consideration, we feel that it has merit but does not fully meet PLOS ONE’s publication criteria as it currently stands. Therefore, we invite you to submit a revised version of the manuscript that addresses the points raised during the review process.

We look forward to receiving your revised manuscript.

Kind regards,

Stuart White, PhD

Academic Editor

PLOS ONE

Additional Editor Comments (if provided):

Dear Dr. Chang,

I apologize for the long review process; I have just accepted this assignment and have reviewed all of the materials up to now.

While reviewer 1 was happy with the revisions, reviewer 2 still had substantial concerns. I have read the manuscript and on balance agree with reviewer 1. There are certainly aspects of the study that could be improved, but that is the case for any study. I feel that the data make a contribution to the field and should be made available.

With that being said, reviewer 2 raised a relevant point with respect to age of sexual debut and adults vs. children. Before publication, I would ask for two additional revisions, both of which should be minor.

1. Please show that the adults in the sample are not suppressing an effect of parenting in the younger part of the sample. You do reference stratifying the sample by age on page 13- however, the results of this stratification are insufficiently addressed in the results section. Whether you chose to re-run the analyses excluding older subjects, better describe the results of including age as a stratification variable, or address this issue via other means is up to you, as long as the fundamental question is addressed.

2. Please address the lack of specific age of sexual debut information as a limitation. It seems highly plausible that sexual behavior in the mid-20's is less related to parenting variables than sexual behaviors in teens. We don't know in the current data, among the older participants, how many participants made their sexual debut in adulthood versus in childhood. I don't feel that this is the fatal flaw that reviewer 2 makes it out to be, but it is worth acknowledging.

If you can address these two points succinctly, I am inclined to accept the paper without further review.

Thank you for your submission and your patience.

Reviewers' comments:

Reviewer's Responses to Questions

**Comments to the Author**

1. If the authors have adequately addressed your comments raised in a previous round of review and you feel that this manuscript is now acceptable for publication, you may indicate that here to bypass the “Comments to the Author” section, enter your conflict of interest statement in the “Confidential to Editor” section, and submit your "Accept" recommendation.

Reviewer #1: All comments have been addressed

Reviewer #2: (No Response)

2. Is the manuscript technically sound, and do the data support the conclusions?

Reviewer #1: Yes

Reviewer #2: Yes

3. Has the statistical analysis been performed appropriately and rigorously? 

Reviewer #1: Yes

Reviewer #2: Yes

4. Have the authors made all data underlying the findings in their manuscript fully available?

Reviewer #1: Yes

Reviewer #2: Yes

5. Is the manuscript presented in an intelligible fashion and written in standard English?

Reviewer #1: Yes

Reviewer #2: Yes

6. Review Comments to the Author

Reviewer #1: The author(s) addressed all of the comments in the previous round of review. Upon reviewing the manuscript a second time, I was particularly impressed with the following:

(1) Formulation of problem statement and significance of the study within the cultural context of Sub-Sahara (not addressed by previous studies in the literature)

(2) Indication that the contextual model of parenting style may not apply to all cultures (more relevant to Western culture)

(3) Excellent use of Baumrind (1960) and Maccoby and Martin (1983) conceptualizations of parenting styles in addition to Darling and Steinberg's contextual model to expand the theoretical framework and literature review for the study

(4) Good delineation of significant gaps in the research

(5) Adequate justification for why students were not randomly selected from classrooms for the two-stage stratified cluster sampling technique (e.g., to increase statistical power).

(6) Good negotiation of entry and access to schools and appropriate procedures for informed consent

(7) Good indication of internal reliability for measures

(8) Good justification for measuring one typology of parenting style (e.g., authoritativeness) based on previous studies

(9) Good indication of inconsistencies of measurement and scoring of parenting style throughout the literature as a weakness and/or limitation for current study

(10) Great recommendations and implications for future research based on significant gaps and conflicting findings in the literature

Reviewer #2: In their response to reviewers, they say they “addressed the reviewers’ comments one by one.” A more common and helpful mode of response to reviewers is to actually list the comments and show how you addressed them.

My first comment, once we’re past the self-plagiarism issue, concerned the age of the sample and the assumption that parental communication will have the same effect on adults as it will adolescents. If this was addressed, it must have been by including age as an iv; however, as I also noted, given that 18 is the age of legal sexual consent, age of debut would have been a better variable. There is little discussion of that but the issue for me is more a question of whether this study is needed. The authors must make an argument that PYSRC even occurs between adults and their parents at a rate that would be effectual in influencing behavior. The lack of finding a relationship with PYSR is thus unsurprising.

Age of debut was not discussed. I'd suggest you offer that as a question for future research.

The paper provides a very nice discussion of cultural context. This section was particularly well done.

The decision, then seems to boil down to whether a paper of non-significant findings makes a contribution. In my opinion, given the fundamental question of adults and parental influence, I do not see enough value, I'm sorry to say. Had the dv been age of debut, perhaps, which is a shame because the paper is very well written.

7. PLOS authors have the option to publish the peer review history of their article (what does this mean?). If published, this will include your full peer review and any attached files.

Reviewer #1: **Yes: **Jewrell Rivers

Reviewer #2: No

---

## [Author Response · Author response to Decision Letter 1]

1 Jan 2021

PONE-D-19-35873

Does parenting style moderate the relationship between parent-youth sexual risk communication and premarital sexual debut among in-school youth in Eswatini?

EDITOR'S COMMENTS: 

Before publication, I would ask for two additional revisions, both of which should be minor.

1. Please show that the adults in the sample are not suppressing an effect of parenting in the younger part of the sample. You do reference stratifying the sample by age on page 13- however, the results of this stratification are insufficiently addressed in the results section. Whether you chose to re-run the analyses excluding older subjects, better describe the results of including age as a stratification variable, or address this issue via other means is up to you, as long as the fundamental question is addressed.

Response:

Thank you so much for the positive comments about our manuscript and for the opportunity you have granted us to revise our work.

In retrospect, we acknowledge that the results from the age-stratified analysis were not adequately addressed in the "Results" section in previous drafts. We now present them in the "Results" section on page 16, lines 366~376, and further discuss them on pages 21 and 22, lines 489~501 in the revised (clean) manuscript. Briefly, we are of the view that since none of the age- and gender-stratified analyses showed statistically significant effects, it does not seem that the adults in the sample were suppressing the effect of parenting in the younger sample, which would have been the case had we found significant effects for the younger sample (<18 years of age). The outputs for all the gender- and age-stratified analyses are found in the S3 SPSS Output file and S4 PROCESS macro output file.-----

2. Please address the lack of specific age of sexual debut information as a limitation. It seems highly plausible that sexual behavior in the mid-20's is less related to parenting variables than sexual behaviors in teens. We don't know in the current data, among the older participants, how many participants made their sexual debut in adulthood versus in childhood. I don't feel that this is the fatal flaw that reviewer 2 makes it out to be, but it is worth acknowledging.

Response:

We actually have information/a variable on age at sexual debut in the dataset, but we did not initially include it in the current analysis as it was not part of the main objective of the current analysis. However, in the revised manuscript, we now include this variable on page 14, line 339 and in Table 1 on page 15. We also cross-tabulated the current age of the youth against their age at sexual debut to ascertain the proportion of adults who had their sexual debut in adulthood versus in childhood. Briefly, the cross-tabulation revealed that nearly 60% of older youth (18~24 years) had their sexual debut in adulthood. We have now discussed this point on page 21, line 492. 

We further performed independent-sample t-tests to ascertain if parenting style and parent-youth sexual risk communication significantly differed by current age of the child and their age at sexual debut, and found no significant mean differences. Results of the cross-tabulations and t-tests are now mentioned in the "Discussion" section on page 22, lines 503~515 and are presented in the supporting information (see S6 SPSS output file).

Having said that, we should point out the limitation in the measurement of the age-at-sexual-debut variable: participants were asked to choose from predetermined categories of 11 or younger, 12~16, and 17 or older, instead of 18 years or older. Unfortunately, that is how the variable is structured in the US Youth Risk Behavior Survey which is the instrument we adopted for our study, and it was deemed appropriate at the time of data collection since the age of sexual consent in Eswatini was 16 years back then. In summary, as much as it is plausible that sexual behavior in older youth is less likely to be related to parenting variables than in younger youth, that does not seem to be the case in the current sample. One main reason could be that Swati parents do not talk to their children about sexuality issues (as discussed in length in the manuscript), independent of the stage or age of the youth, such that the slope is constant across the adolescent or youth life course. We have presented the above points on page 22. We hope that the information above provides clarity to the issue of parenting and older youth.

REVIEWER COMMENTS:

Reviewer: 1

Comments: 

The author(s) addressed all of the comments in the previous round of review. Upon reviewing the manuscript, a second time, I was particularly impressed with the following: (list of comments omitted)

Response:

Thank you for the positive feedback regarding our work.

Reviewer: 2

Comments: 

1. The paper provides a very nice discussion of cultural context. This section was particularly well done.

Response: Thank you for the positive feedback.

2. In their response to reviewers, they say they “addressed the reviewers’ comments one by one.” A more common and helpful mode of response to reviewers is to actually list the comments and show how you addressed them.

Response:

We actually did that in our first revision. We listed each comment from both reviewers in a table with two columns: one with the comments and the other stating how each comment was addressed.

3. Age of debut was not discussed. I'd suggest you offer that as a question for future research. The decision, then seems to boil down to whether a paper of non-significant findings makes a contribution. In my opinion, given the fundamental question of adults and parental influence, I do not see enough value, I'm sorry to say. Had the dv been age of debut, perhaps, which is a shame because the paper is very well written.

Response:

We actually have information/a variable on age at sexual debut in the dataset, but we did not initially include it in the current analysis as it was not part of the main objective of the current analysis. However, in the revised manuscript, we now include this variable on page 14, line 339 and in Table 1 on page 15. We also cross-tabulated the current age of the youth against their age at sexual debut to ascertain the proportion of adults who had their sexual debut in adulthood versus in childhood. Briefly, the cross-tabulation revealed that nearly 60% of older youth (18~24 years) had their sexual debut in adulthood. We have now discussed this point on page 21, line 492. 

We further performed independent-sample t-tests to ascertain if parenting style and parent-youth sexual risk communication significantly differed by current age of the child and their age at sexual debut, and found no significant mean differences. Results of the cross-tabulations and t-tests are now mentioned in the "Discussion" section on page 22, lines 503~515 and are presented in the supporting information (see S6 SPSS output file).

Having said that, we should point out the limitation in the measurement of the age-at-sexual-debut variable: participants were asked to choose from predetermined categories of 11 or younger, 12~16, and 17 or older, instead of 18 years or older. Unfortunately, that is how the variable is structured in the US Youth Risk Behavior Survey which is the instrument we adopted for our study, and it was deemed appropriate at the time of data collection since the age of sexual consent in Eswatini was 16 years back then. In summary, as much as it is plausible that sexual behavior in older youth is less likely to be related to parenting variables than in younger youth, that does not seem to be the case in the current sample. One main reason could be that Swati parents do not talk to their children about sexuality issues (as discussed in length in the manuscript), independent of the stage or age of the youth, such that the slope is constant across the adolescent or youth life course. We have presented the above points on page 22.

---

## [Editor Report · Decision Letter 2]

5 Jan 2021

Does parenting style moderate the relationship between parent-youth sexual risk communication and premarital sexual debut among in-school youth in Eswatini?

PONE-D-19-35873R2

Dear Dr. Chang,

We’re pleased to inform you that your manuscript has been judged scientifically suitable for publication and will be formally accepted for publication once it meets all outstanding technical requirements.

Kind regards,

Stuart White, PhD

Academic Editor

PLOS ONE
---

## [Editor Report · Acceptance letter]

11 Jan 2021

PONE-D-19-35873R2 

Does parenting style moderate the relationship between parent-youth sexual risk communication and premarital sexual debut among in-school youth in Eswatini? 

Dear Dr. Chang:

I'm pleased to inform you that your manuscript has been deemed suitable for publication in PLOS ONE. Congratulations! Your manuscript is now with our production department. 

Kind regards, 

on behalf of

Dr. Stuart White 

Academic Editor

PLOS ONE